# Recent Developments in Haptic Devices Designed for Hearing-Impaired People: A Literature Review

**DOI:** 10.3390/s23062968

**Published:** 2023-03-09

**Authors:** Alejandro Flores Ramones, Marta Sylvia del-Rio-Guerra

**Affiliations:** Computer and Industrial Engineering Department, Universidad de Monterrey, San Pedro Garza Garcia 66238, Nuevo Leon, Mexico

**Keywords:** haptic devices, hearing impairment, human–computer interaction, wearable devices, sensory substitution systems, vibrotactile feedback

## Abstract

Haptic devices transmit information to the user, using tactile stimuli to augment or replace sensory input. People with limited sensory abilities, such as vision or hearing can receive supplementary information by relying on them. This review analyses recent developments in haptic devices for deaf and hard-of-hearing individuals by extracting the most relevant information from each of the selected papers. The process of finding relevant literature is detailed using the PRISMA guidelines for literature reviews. In this review, the devices are categorized to better understand the review topic. The categorization results have highlighted several areas of future research into haptic devices for hearing-impaired users. We believe this review may be useful to researchers interested in haptic devices, assistive technologies, and human–computer interaction.

## 1. Introduction

Based on the number of people with a hearing loss greater than 35 decibels in their better ear [1], the World Health Organization has estimated that more than 5% of the world’s population (around 430 million people) requires rehabilitation or treatment to deal with hearing impairment. However, this estimate increases to 1.5 billion when the number of people with a noticeable level of hearing is also taken into account [2]. In addition, it is estimated that this number could reach 2.5 billion people by 2050 [1].

Thus, given the high number of hearing-impaired people, efforts aimed at supporting the said population can positively impact a large part of the population, including those close to them. For example, parents of children with disabilities who receive support and have an action plan experience lower stress levels [3].

The World Health Organization has also estimated that the annual cost to the global economy due to the lack of proper care of the hearing impaired is 980 billion USD. These costs are transferred to the health sector (not including the cost of hearing devices such as cochlear implants), educational support costs, reduced productivity, and social costs [1]. In other words, efforts directed toward addressing the disability would not only benefit the concerned people but also their families and friends, as well as the society at large.

As Julia Taylor and Kellie Mote point out, accessibility refers to designing systems that allow everyone the same access and opportunities, regardless of their personal characteristics, ensuring that all can fully participate [4]. Therefore, accessibility is a step towards inclusion.

Certain technology tools for assistance can help people with hearing disabilities, thereby improving their accessibility and inclusion. One such example is the haptic technology, which consists of devices that apply tactile stimulation to the user. This technology allows deaf and hard-of-hearing (DHH) individuals to receive information through one of their fully functioning senses.

Aside from technology, policy implementation can also contribute to inclusion. In this regard, the United Nations has established a strategy titled “United Nations Disability Inclusion Strategy”. This strategy has four areas: leadership, strategic planning and management, inclusion, programming, and institutional culture [5].

The strategy aims to collaborate with people with disabilities in order to create an inclusive environment where they can integrate with both internal work of the parent organization and the external work of other organizations.

Haptic devices and technology play a key role in achieving the above strategy. In order to understand the relevance, it is important to first to know the difference between equality and equity. Paula Dressel states: “The path to equity will not be met by treating everyone equally. It will be achieved by treating everyone fairly according to their circumstances” [6].

This means that if you treat everyone equally, then individuals in disadvantaged situations will still be disadvantaged. For example, if a company offers video calling tools to all its employees to promote online meetings, a deaf employee will still be unable to hear their colleagues during such meetings, unless they have the option to transcribe the audio. Thus, to promote fairness, a deaf employee should receive support through a transcription tool.

Technology plays an important role in inclusion. It enables people with disabilities to fully participate in activities where they might otherwise be excluded from. People with partial or complete loss of one or more senses can better integrate into different activities by using devices that serve as sensory augmentations or substitutions, such as haptic devices.

In a previous study [7] the authors illustrated the potential of technology in promoting inclusion. Water polo is a sport played in a swimming pool, so athletes using hearing aids need to remove the device. As a result, they cannot hear the referee’s whistle during a match and have slower reaction times, which negatively impacts their team’s performance. To solve this problem, a waterproof haptic device has been developed that detects the sound of the whistle and translates it into vibrations in the athlete’s ear. If this device is available, offered and supported, water polo can become a more inclusive sport for hearing-impaired players.

People with disabilities face a variety of challenges that differ from those faced by the rest of the population. Technologies, including those mentioned above, are not intended to solve the entirety of their barriers. However, these devices can still be of substantial help in reducing the problems experienced during certain activities.

The Web Accessibility Initiative mentions three relevant aspects of accessibility [8]. First, it emphasizes that accessibility benefits people with or without disabilities. Second, it indicates that accessibility features are increasingly available in standard computing hardware, mobile devices, operating systems, web browsers, and various other tools. Lastly, it highlights that people perceive the world in a variety of ways, including touch, which is directly related to haptic technology. In addition, the Web Accessibility Initiative specifies that an accessible tool should be perceivable, operable, understandable, and robust [9]. Haptic devices provide increased opportunities to people with different disabilities by using touch as a communication channel, which differs from traditional media. For example, a device could increase its perceptibility by implementing tactile feedback. Similarly, devices that offers several options to communicate the same information, including the haptic channel, become more robust.

### 1.1. Hearing Impairment

Hearing impairment generally refers to a diminished ability to hear sounds and can range from barely perceptible noise to total deafness. It can originate from different areas, such as the conduction of sound to the inner ear, the perception of sound by the sensory cells of the cochlea, the processing of sound by the cochlear nerves, the auditory pathway, or cortical auditory centres [10].

Hearing impairment or hearing loss should be distinguished from other hearing disturbances such as hypersensitivity to sound (hyperacusis) or tinnitus, although these conditions can be caused by auditory loss itself [11].

Hearing loss implies that an individual is unable to hear as well as the average person, which is at least 20 decibels in both ears. It can be classified as, depending on its severity, into mild, moderate, moderately severe, severe, or profound hearing impairment [1].

Hearing impairment can be described through a variety of aspects. Depending on whether it affects one or both ears, hearing impairment can be unilateral or bilateral. Based on whether the person had already learned to speak before the hearing loss, hearing impairment can be pre-lingual or post-lingual.

In addition, hearing impairment can be symmetric or asymmetric depending on whether the hearing loss is the same in each ear. Moreover, it can be classified as either progressive depending on whether the loss increases over time or not. If the hearing loss is present at birth, it is said to be congenital, and if it appears at a later age, it is considered to be acquired [12]. From another standpoint, conductive hearing loss is present if sound conduction is impeded in some way through the external ear, the middle ear, or both. Sensorineural hearing loss is present if there is a problem with the cochlea or with the cochlea or neural pathway to the auditory cortex. Mixed hearing loss means both conductive and sensorineural loss are present.

Hearing loss can be caused by a variety of factors. On the one hand, hearing loss can be caused by temporary problems, such as wax build-up in the outer ear. Hearing loss due to eardrum ruptures caused by objects inserted into the ear, pressure, or sudden loud noises can heal over time if the loss is not severe. Ototoxic medicines, which can damage the ear, can result in temporary or permanent damage to the ears. On the other hand, constant exposure to loud sounds that damage the inner ear, as well as certain congenital or postnatal infections can cause permanent hearing loss. In addition, hearing loss can be hereditary or caused by conditions such as diabetes, high blood pressure, heart attacks, brain damage, or tumours [13].

### 1.2. Haptic Devices

Haptic devices allow for human–computer interaction through touch and external forces. Unlike traditional interfaces, such as visual screens and audio systems, haptic devices present mechanical signals that are perceived by human touch [14].

Haptic devices focus on tactile communication to transmit information, and these forms of information communication include the use of pressure, vibrations, and temperature [15]. In general, haptic devices work with the skin (the body’s largest sensory organ), the musculoskeletal system, and other tissues. They send perceptual signals in the form of forces, displacements, or through electrical or thermal inputs to the user’s skin and body. Specifically, the skin plays a critical role in perceiving and interacting with the environment [16]. Moreover, touch is an important aspect for human development [15].

Fleury et al. [17] present three general classifications of haptic devices. First, haptic devices can either be portable or anchored to the environment. They can also involve passive or active touch. Passive touch refers to the physical property of touching an object without exploring it; active touch refers to physically exploring an object through touch. Active touch is generated by the device using actuators and software, as most haptic devices use. Finally, the haptic device can either be in direct contact with the user at all times, intermittently, or be in indirect contact, as is the case with ultrasound.

Haptic devices can be further classified based on the type of their tactile interaction. As described in a previous study [18], there are a variety of tactile interactions, which are summarized below. Currently, there are no formal haptic device categories that correspond to different tactile interactions. Nevertheless, they can still serve as a broad classification tool.
Vibration: applied normally or transversely to the skin’s surface and can vary in frequency, amplitude, duration, or timbre.Contact/Pressure: an object can make contact with the body and then break that contact, alternating between these two states or varying the level of pressure.Temperature: changes in temperature, such as going from cold to hot, or from hot to even hotter.Geometry: shape, such as the curves or reliefs of an object.Texture: the sensation generated by touching certain surfaces, which may be smooth or rough, for example.Softness/Hardness: a material’s resistance to pressure, such as when pressed by a finger.Electricity: electrical stimuli can be used to generate haptic sensations, known as electro-haptic stimuli.Friction: resistance to movement between two solid bodies whose surfaces are in contact.


### 1.3. Use of Haptic Devices for the Hearing Impaired

Sensory augmentation, as part of “hybrid biology”, involves the addition of synthesized information to a person’s sensory channel [19]. An example of sensory augmentation includes haptic devices, which transmit information through touch [20]. When external factors impact the information perceived through senses, people can use sensory augmentation. An example of this is the device shown in [21]; this device consists of a helmet for firefighters that uses vibrations to help them navigate in low-visibility areas.

Sensory substitution is a similar concept that is described as the use of one sense to provide environmental information that can normally be obtained from another sense [22]. Haptic devices can also be used here, as demonstrated in [23]. These devices use vibrations to alert people with hearing impairment about sounds coming from behind, such as greetings or calls for attention.

Compared to visual feedback, haptic feedback does not interfere with the user’s view, and can be applied anywhere on the body [24]. As a result, haptic stimuli are considered to be relatively discrete. The haptic channel does not involve the user’s vision and may be useful when someone is watching a movie in a foreign language. That individual either sees the movie or reads the captioning.

These examples demonstrate how haptic devices are capable of augmenting and substituting sensory information, while this literature review focuses on hearing impairment, haptic devices can support the other senses, as compiled by [25]. These features provide an overview of the potential of haptic devices that can benefit people with different disabilities in a variety of contexts.

## 2. Previous Literature Reviews

There are six main previous literature reviews as shown in Table 1. Each one is explained in greater detail.

Shull and Damian [25] focused on the other senses of the human body, besides hearing impairment. Their paper examined the use of haptic devices in supporting people with hearing impairment and enhancing the capabilities of people in general. In addition, this review covers only “wearable” devices. In comparison, the present paper aims to cover haptic devices for hearing impairment in greater detail and show the recent developments in the area.

Sorgini et al. [20] focused on haptic devices that help people with hearing or vision impairments. As a review with a narrower scope than the paper by Shull and Damina [25], this paper covered more details about haptic devices for hearing impairments. However, as it was published in 2017, it does not cover the most recent developments in the area. Therefore, there is a need to update the overview of this area by reviewing the literature published since then.

In a recent paper by Fletcher [26], haptic stimuli were used in conjunction with today’s traditional assistive listening devices, such as cochlear implants in order to improve sound separation and localization. Fletcher noted that users of such devices often struggle with these two aspects. Thus, in this paper Fletcher reviewed papers with a medical approach to derive a rationale for using haptic stimuli in conjunction with cochlear implants. This paper, however, did not describe the different haptic devices that can benefit people with hearing impairments.

Two other literature reviews cover the topic of haptic stimuli as a replacement for sound but, specifically, for the purpose of communicating musical information. The paper by Remache et al. [27] compiled different methods and technologies used to communicate music through touch. Meanwhile, another paper Fletcher [28] discussed studies showing that haptic stimulation can improve musical perception in people with hearing impairments.

There are also survey articles on haptic stimulation such as [29]. This review focused on haptic interfaces, but hearing disabilities were only mentioned in passing. Thus, there is little relation with the focus of our review. Table 1 lists the focus of each of these reviews. Since we have only considered those that specifically mention hearing devices with haptic actuators, the paper [29] was not included.

In conclusion, other reviews focus on particular aspects of hearing impairment: music, localization, and cochlear implants. No reviews are dated after 2021, and one paper does not focus on accessibility issues.

## 3. Review Methodology

This study presents a review of haptic devices with applications for hearing impairment In addition, possible areas of opportunity and future work were discussed. The study followed an adaptation of the PRISMA guidelines [30].

### 3.1. Research Question

The following research question guided the literature review, including the search and selection process.

What have been the recent developments in haptic devices aimed towards helping people with hearing impairments?

### 3.2. Search Strategy

Different electronic databases and digital libraries were searched to locate relevant published papers and reduce any bias that may arise from using a single source, as seen in Table 2. Multiple searches were conducted during the review process to ensure inclusion of recently published content.

In order to construct the final search strings, terms related to hearing impairment were grouped into one category, while terms related to haptic devices were grouped into another category. The basic terms related to hearing impairment were:Deaf;Hard-of-hearing;DHH.

As well as the combination of the following two lists of terms:Hearing, auditory, audition;Impairment, loss, deficiency, disability, problem, impediment.

The basic terms related to haptic devices were:Haptic;Vibrotactile;Electrotactile.

As well as the combination of the following two lists of terms:Tactile, tactual, touch, vibration;Aid, device, technology, display, interface, feedback.

Terms from each category were combined using an "AND" operator to search for potential papers relating to haptic devices for hearing-impaired individuals. Proximity operators were used for the last items in each category, which consist of a list of terms that should be combined with another list of terms. Thus, the following general search string was generated (with syntactic variations for different sources):


(*haptic* OR ((tactile OR tactual OR touch OR vibration) PRE/5 (aid OR device OR technology OR display OR interface OR feedback)) OR vibrotactile OR electrotactile) AND (((hearing OR auditory OR audition) PRE/5 (impair* OR loss OR deficiency OR disability OR problem OR impediment)) OR deaf* OR DHH OR "hard-of-hearing")


Some sources did not offer proximity operators. To obtain the different possible combinations for these cases, another search string was created by joining the previously mentioned pair of term lists with a script. This search string was:


(haptic* OR "tactile aid" OR "tactile device" OR "tactile technology" OR "tactile display" OR "tactile interface" OR "tactile feedback" OR "tactual aid" OR "tactual device" OR "tactual technology" OR "tactual display" OR "tactual interface" OR "tactual feedback" OR "touch aid" OR "touch device" OR "touch technology" OR "touch display" OR "touch interface" OR "touch feedback" OR "vibration aid" OR "vibration device" OR "vibration technology" OR "vibration display" OR "vibration interface" OR "vibration feedback" OR vibrotactile OR electrotactile) AND ("hearing impairment" OR "hearing loss" OR "hearing deficiency" OR "hearing disability" OR "hearing problem" OR "hearing impediment" OR "auditory impairment" OR "auditory loss" OR "auditory deficiency" OR "auditory disability" OR "auditory problem" OR "auditory impediment" OR "audition impairment" OR "audition loss" OR "audition deficiency" OR "audition disability" OR "audition problem" OR "audition impediment" OR deaf* OR DHH OR "hard-of-hearing")


The full information about the search strings used for each source is shown in Appendix A.

### 3.3. Inclusion and Exclusion Criteria

A set of inclusion and exclusion criteria were established to determine which papers should be selected. There were two types of criteria: general criteria (which could be applied to any paper) and content-specific criteria (specific to this review’s topic). The general inclusion criteria were as follows:The paper should be published between 2017 and 2022 (inclusive);The paper should be published in English;The full text of the paper should be available online or be accessible by contacting the corresponding author;The document should be either a journal article or a conference paper.

The content-specific inclusion criteria were as follows:The paper should explicitly indicate that the developed device is aimed at people with hearing impairments.

The general exclusion criteria were:Papers published in non peer-reviewed sources (also known as “grey literature”) were not considered;Literature reviews were not considered.

The content-specific exclusion criteria were:Papers that described devices without developing their own were not considered;Papers about devices that only included a minimal haptic feature which was not its main focus, were not considered.

### 3.4. Quality Assessment

In addition to the inclusion and exclusion criteria, three more aspects were considered as a proxy to assess the quality of the relevant papers for the review. The aspects were as follows:The paper should present a developed prototype/device that goes beyond the design or conceptual stage;The device should have been tested by at least one user;The paper should include the results of the tests.

### 3.5. Study Selection

The search results from all the sources, excluding Google Scholar, provided 1745 papers. Google Scholar returned 18,700 results. Of these, only the first 53 pages were considered (the selection process ended 10 pages after the last relevant result). As Google Scholar displays 10 elements per page, 530 records were counted for this source to be added to the 1745 number. In total, 2275 results were found. The content-related inclusion and exclusion criteria were not considered when selecting papers from the search results. At this stage, the only content-related criterion was selecting papers related to haptic devices with applications for hearing impairments. This allowed to focus on narrow selection criteria when working with many papers. Consequently, we could perform the process more effectively and with less errors.

In addition, it reduced the number of false negatives. Later, the content-related inclusion and exclusion criteria were applied to the reduced number of papers to further refine the initial selection.

During the first stage of the selection process, which consisted of going through the search results, the only content-specific criterion considered was selecting papers related to the topic of haptic devices with applications for hearing impairments. At this stage, the number of papers is at its highest, and this simpler selection criterion allowed us to be more effective and less error-prone. In addition, the criterion reduced the number of false negatives. The initial selection was improved further using the entirety of the inclusion and exclusion criteria.

The following steps were taken to determine whether these papers passed the inclusion and exclusion criteria:Title inspection;Abstract inspection;If the paper was not discarded from the previous criteria, its contents are inspected.

A total of 25 papers were finally included in the literature review.

Figure 1 shows the PRISMA flow diagram for paper selection. A total of 2275 results were obtained by searching across the different databases. The number of results reduced to 342 when only the papers relevant to the review topic were selected. After removing duplicates that appeared in different databases, the number of papers was reduced to 285. After the application of the inclusion and exclusion criteria, 32 papers remained. Finally, a quality assessment of the papers resulted in 27 papers.

## 4. Literature Review Results

Figure 2 shows the years in which the selected papers were published. The number of papers seems to be similar throughout the years, with greater numbers in the years 2020 and 2021, although the range may not be long enough to extract a trend. These relatively high fluctuations can be expected, such as the year 2020 publishing double the amount of papers as the previous year, due to the low number of papers that resulted from the inclusion and exclusion criteria relevant to this literature review.

Figure 3 shows the number of selected papers from each country. The United States topped the list with six papers, followed by Japan with five papers. Following that, Denmark, South Korea, and Turkey had two papers each. The rest of the countries had one paper each.

The different body parts where haptic stimuli were applied by the devices from the 25 selected papers are shown in Table 3. The middle column in the table sums to 35, which is more than the 25 selected papers. This is because some devices were designed to be worn in multiple areas of the body, or were tested in different places to compare the performance results.

Fourteen of the 35 locations involved the wrist, hand (back and/or palm), and fingers. The fingers and hands were the most commonly used body area for haptic stimuli application, as five papers included each of them. Six papers showed haptic stimulation applied to the arms, divided among the upper arms and forearms. Three papers involved the legs, including the upper and lower legs. Apart from the back and chest, which appeared in two papers, all of the other body parts appeared only once. Figure 4 graphically illustrates which body parts are most commonly used by researchers.

As shown in a previous study [37], two adjustable straps with haptic actuators were used to allow users to choose where the haptic stimuli were delivered. In a similar study [38], a haptic device was developed in two different formats. One design was to be worn as a glove and the other as a belt.

In two instances the haptic stimuli were not limited to a single location, but rather the entire body experienced the stimuli to varying degrees. In [31], the user laid down on a vibrating bench, and in [32], a robot hit the bed where the user was sleeping causing it to vibrate. In another study [53], no defined body part received the stimulation. In this case, the vibrations were generated by a smartphone, which could be in various locations depending on the circumstances.

As shown in Table 4, the haptic devices in the 25 selected papers were used in a different number of locations for stimuli application. Note that this is not the number of distinct body parts where haptic stimulation was applied. For example, in [40], the 12 vibration motors used were positioned on the forearm, and each motor sent haptic stimuli to a different specific location. In addition, this metric did not necessarily correspond to the number of actuators in the device. As an example, if multiple motors vibrate a single piece that is in contact with the user, it would be counted as one stimulation location. In total, 8 of the 25 papers (32%) used a single location for simulation.

Regarding the reason for using a single stimuli location, the information conveyed through the haptic stimuli does not rely on specific patterns in cases such as [32,53]. Instead, the device simply transitions from no haptic stimulation to then applying said stimulus, similar to an alert. Other devices, as in [45], transmit more complex information through a single location. For this, they used vibrations of different magnitudes to represent different sounds. In addition, a single location for haptic stimuli could convey complex information by utilizing concepts such as Morse code. The stimulus would alternate between on and off periods for different lengths of time in order to represent patterns.

Some devices, such as in [34,36], use multiple stimulus locations to guide the user. For this, the locations to be activated are changed, and their intensities are adjusted in relation to one another. The device in [40] also employed multiple locations, each corresponding to a different pitch. Although a single stimulus location can transmit all the information that multiple locations can, practicality and usability concerns may make the alternative more appealing to researchers and users. Alternatively, a device using a single location for stimuli could be redesigned to employ multiple locations, but it may not provide any advantage.

As shown in Table 5, 7 (28%) out of the 25 haptic devices were used to alert users. Sound can alert people to situations that might otherwise go unnoticed and result in negative outcomes. For example, when walking on the street, you can avoid being hit by a vehicle coming from behind if you hear honking and screeching tires. This example was used in both [23,51]. Thus, devices that use tactile stimuli to alert DHH individuals are of significance. It is important that individuals with hearing loss/impairment be aware of such sounds because they typically require immediate attention and action. Therefore, haptic devices that focus on alerts are expected to become more prevalent.

In six papers, the haptic devices used helped the users to experience music. These devices used vibrations to represent or complement music, whether to enhance the experience for hard-of-hearing users, or to allow deaf users to experience it in their own way. This category of devices may be of greater interest to researchers due to the prevalence of music in history and culture.

As shown in Table 6, haptic devices use a variety of actuator types. Actuators are components that generate haptic stimulus. In most of the selected studies, vibrating components such as resonant linear actuators or eccentric rotating mass motors, were used. These vibrating components are either in direct contact with the user, or with another element that then transmits the vibration to the user. Even though smartphone vibration is caused by a motor within the phone, it was included in its own category because its presentation stands out from the rest. All of the actuators, except for the bumping robot presented in the last row of the table, use a vibrating component.

Table 7 shows that 4 of the 25 haptic devices included in the selected papers were not portable, while the remaining 21 were portable. Two of these non-portable devices were considered as furniture: one was a bench where people laid down to feel vibrations [31], and the other was a chair with speakers and subwoofers [35]. Another non-portable device used was a set of speakers that vibrated a table [46] so that the user felt the stimuli. The fourth non-portable device was a bumping robot in [32] that hit a bed to alert the user.

Table 8, focuses on haptic device specifications. The device receives sound input, which is translated into haptic stimuli. Devices that do not gather audio or translate sounds directly may not always contain input acquisition. The table also shows whether audio processing was applied and how haptic stimuli are applied to the user. It also shows the number of people who tested the device.

As shown in Table 8, the majority of the devices gather sound from the environment using microphones, while a minority use prepared audio files. The most variations among the devices were in terms of the number of stimuli application zones and signal processing. In general, audio input was translated to haptic stimuli by extracting key features and applying a few trigger points. All these devices used vibration as a method of applying stimuli. In most cases, the vibration devices were portable and were used either directly on the user or integrated into wearables such as gloves or wristbands.

## 5. Discussion

Our results showed that fingers and hands were the most popular body parts for haptic stimulation because they are highly sensitive to touch, making them a convenient choice for haptic stimulation. In addition, people generally use hands and finger for touching and interacting with their physical environment. Another key factor to consider is how practical or common the devices are for those parts of the body. Bracelets allow for the application of haptic stimuli to the wrist, and people are accustomed to such designs because of other wearables such as wristwatches. Gloves can also be used to apply haptic stimuli to the wrist. Users may find it more familiar and comfortable wearing technology in these forms. In addition, haptic actuators can be easily incorporated into them. It is important to mention that despite the many advantages of haptic actuators on the hands, this reduces the user’s ability to employ them for other things.

Note that neither the face nor the feet were targeted for haptic stimulation. The face is a crucial part of one’s identity and communication, and one may not want the haptic device to obstruct it. In addition, some may find haptic stimuli to the face to be more inconvenient than placed elsewhere. The feet, are usually covered by footwear, and it may not be practical to place a device on them. Similar to wristwatches and wristbands, haptic devices can be placed on the ankles. However, the sensitivity of ankles must be tested in order to determine their usefulness.

The concept of using haptic devices to apply haptic stimuli to different body parts can be explored further. From among our selected studies, only a few devices used this concept. The device shown in [34] used the chest, upper back, and thighs in one of its designs. In addition, the device from [37] applied stimulation to the upper arm and chest. There were, however, some devices that applied stimuli to both the right and left sides of a body part. For example, [36] used the right and left shoulders, while [52] used the right and left legs.

In the selected papers haptic devices that allow for real-time interaction between multiple people were not directly represented. Some papers, however, indirectly incorporated the concept. The haptic device shown in [23] alerted users to sounds coming from behind, and its tests included the recognition of greetings and attention calls, which relate to communication. Moreover, ref. [40] presented a haptic device that improved pitch perception for cochlear implant users and further enhanced speech recognition. Finally, the device in [39] helped the user to learn English words, also related to communication. Nevertheless, the main focus in the previous examples was not on interacting with others. People with hearing impairment/loss can communicate through sign language, so haptic devices focused on communication may not be a priority. In contrast, deaf and blind individuals may benefit more from such devices, as evidenced in papers [54,55], which were excluded from this review due to their target population (multiple impairments). Nevertheless, haptic devices can involve multiple participants. Among the 25 reviewed devices, none were intended to be used concurrently by more than one person. Multiple-user devices could, for example, be a single physical entity that people interact with simultaneously, or multiple copies of the same device that people use to interact with each other.

Haptic devices that enable DHH people to create content rather than consume or experience is another aspect that can be explored. Such devices will allow users to actively participate in interactions. In a closely related paper [24], a device was designed to improve pitch perception in cochlear implant users, with a particular focus on singing. For example, in [37], a deaf participant took part in co-creating music–movement performances. Similarly, the authors in [38] developed a device to help users learn dance performances. However, the focus was on learning a routine rather than on individuals making their own. The small number of studies of this nature may be due to the desire to address more “basic” problems and/or problems lower in the hierarchy of human needs. However, more devices that put users in an active role may emerge with further development in the field of haptic devices.

Haptic devices tailored to a particular industry or profession could be developed. The majority of the reviewed devices are suitable for a wide range of users. DHH individuals from diverse backgrounds can benefit from using haptic devices intended for experiencing music, or devices that alert people to critical sounds in their homes. Although such devices are useful for a large percentage of the DHH population, developing devices that help in more niche areas may also be worthwhile. This could include devices for specific professions, as mentioned before. This will enable more industries to become more inclusive and accessible. A device using this concept was described in [50], which was designed to assist DHH students in learning how to use drawing software. With the growth in industries, research could branch out into more specific areas and better cover general-purpose devices.

The bumping alert robot shown in [32] was the only device without a vibration element. The robot hits a person’s bed so they can feel the haptic stimulus and be alerted. This allowed a person to feel vibrations when the robot hits the bed even if the bumping robot does not include a vibrating actuator. In other words, all the haptic stimuli employed by the devices in the review are vibrations. Thus, devices that utilize actuators other than vibrating components or employ stimuli other than vibrations are novel research opportunities. Despite this, vibrating actuators are inexpensive, easy to obtain, and have practical uses, of which they are expected to be very popular. In general, it is fairly simple to attach a coin-shaped vibration motor to something such as a glove that the user wears. Other types of stimuli, such as thermal, pressure-based, or electro-haptic, could be further explored.

Due to the limited number of portable devices, there is scope for further exploration. Given that wearable haptic devices are easy to use, it is expected that the number of these devices will increase tremendously. However, none of the studied devices have combined mobile and stationary components. For example, considering a hypothetical device with portable components that can be used alone and be coupled with non-portable components for a different experience.

Lastly, the selected papers did not explore the concept of configurable haptic devices where users could customize their behaviour. A device that uses vibrations might allow users to adjust their intensity. Devices that detect sounds in the environment and map them to haptic stimuli, such as the one in [43], might allow users to adjust the detection threshold. Devices, such as in [45], which map sounds to vibration patterns, might allow the user to select which sounds to map and which ones to ignore. In addition, such devices may allow the user to select which vibration patterns to use. Like many devices and technologies, haptic devices could also offer a customized experience by allowing users to configure them as per their requirements. This is especially important for devices that are designed to assist people with disabilities, who face unique challenges. The selected papers did not take this approach, as the devices presented are considered to be proof-of-concept, meant to be refined and expanded upon. Moreover, a device’s settings should be maintained throughout a study in order to accurately examine it, restricting the configuration options.

## 6. Conclusions

This study reviewed recent peer-reviewed literature on haptic devices for DHH people. The selected studies showed that haptic devices exhibit extraordinarily varied aspects, while some other aspects, mostly related to hardware and technical features, much less so. Researchers might find it interesting to explore these lesser-represented topics in future studies, even though this is not necessarily a key issue.

The haptic devices showed the most variations with respect to the number of stimulation zones, the area where the stimulation was applied, and the purpose of the device. Nevertheless, there were prevailing categories within each of these criteria. Most devices used a single location for the stimuli. Hand and finger stimulations were the most common application zones. Most of the devices were wearable and focused on alerts. Minimal variety was observed in terms of portability, general actuator type, or haptic stimulation type. Moreover, the majority of actuators were vibration motors. Finally, vibration was the only haptic stimulation method used.

Placement of devices at other areas, such as those previously described, may receive more attention as the haptic device research area matures. Future research should focus on actuators on different body parts. If actuators can be used on the arms, legs, or chests, it will be useful for people without arms, for example. Further research must be performed on electrical and heat stimuli. Currently, the under-represented haptic devices may not be as convenient or practical as other alternatives, but this may be improved with technological advancements. It is also possible to use artificial intelligence in order to send alerts when a user needs to pay extra attention to sound stimuli by combining cochlear implants with haptic devices. Given that the global hearing-impaired population is expected to reach 2.5 billion by 2050 [1], devices that promote accessibility and inclusion can positively impact the lives of many people.

## Figures and Tables

**Figure 1 sensors-23-02968-f001:**
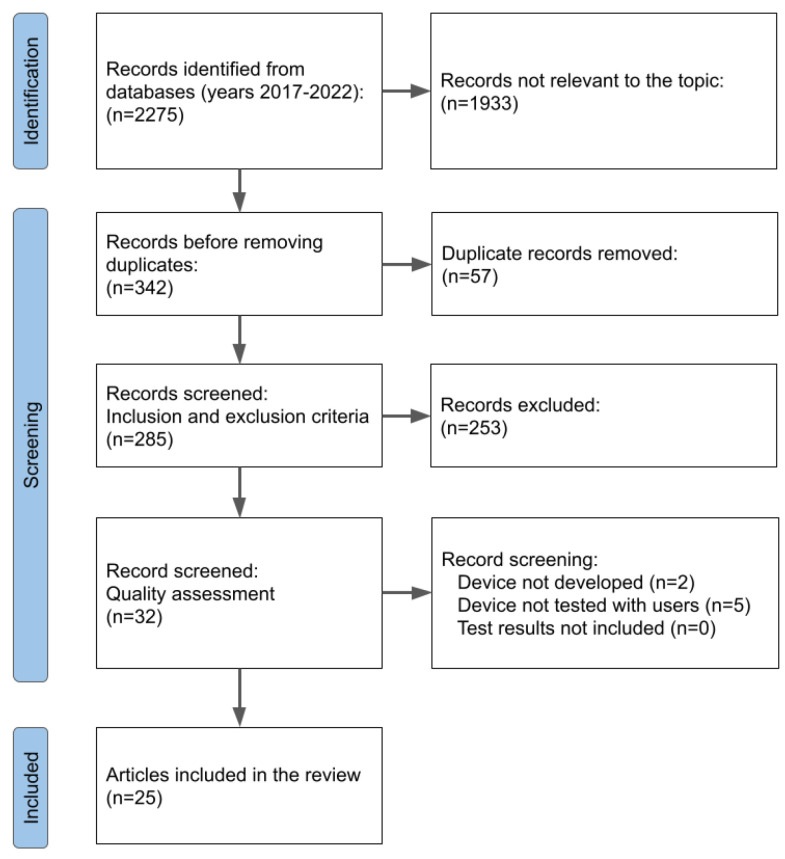
PRISMA flow diagram for the review process displaying the number of excluded papers for each stage.

**Figure 2 sensors-23-02968-f002:**
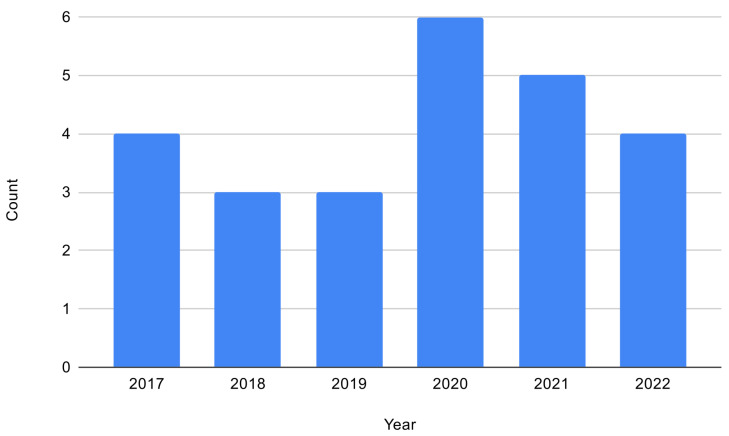
Papers by year of publication.

**Figure 3 sensors-23-02968-f003:**
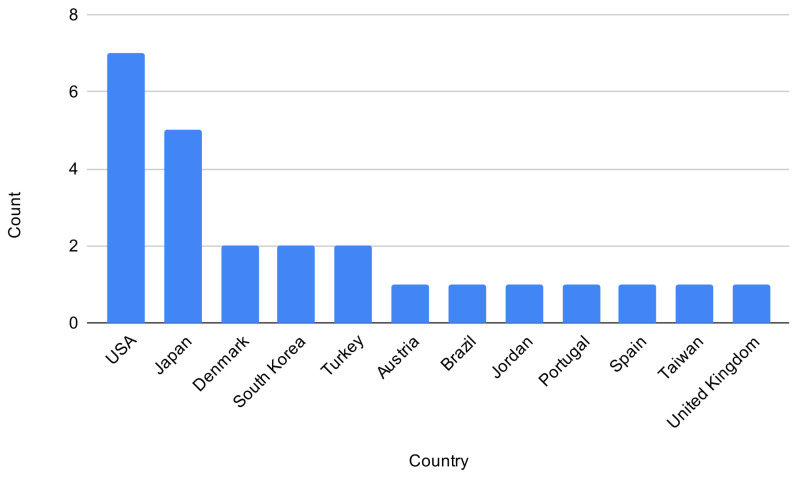
Countries by number of published papers.

**Figure 4 sensors-23-02968-f004:**
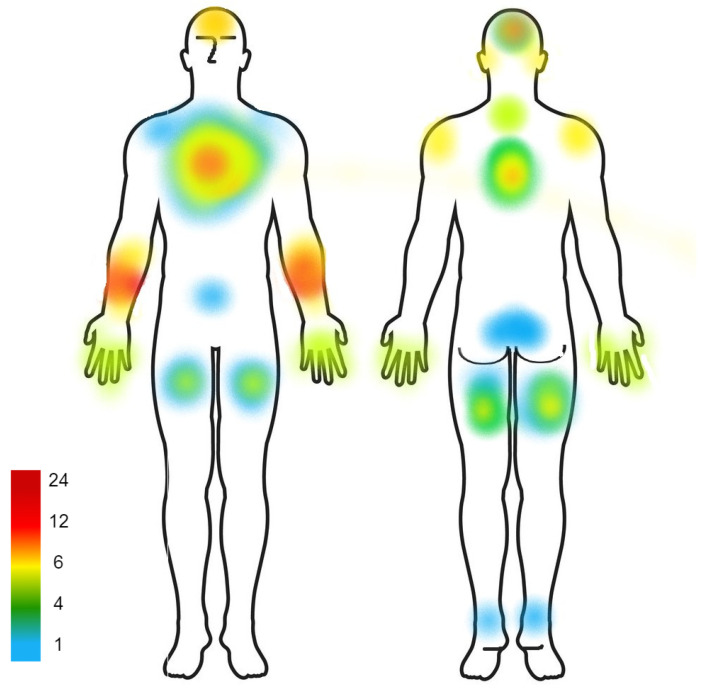
Commonly used body parts.

**Table 1 sensors-23-02968-t001:** Other literature review papers.

Focus	Devices	Authors	Papers in Review
Total or partial impairment of different sensorial disabilities	Wearable devices	[25]	9
Several senses	Vision and hearing	[20]	20
Sound separation and localization	Cochlear implants	[26]	16
Music	Technology to transmit music	[27]	15
Music	Haptics to improve music perception	[28]	21

**Table 2 sensors-23-02968-t002:** Electronic databases and digital libraries searched for the literature review process.

Source	URL	Date of Last Search	Results
Web of Science	http://webofknowledge.com/	19 December 2022	255
Scopus	https://www.scopus.com/	19 December 2022	208
Wiley Online Library	https://onlinelibrary.wiley.com/	18 December 2022	338
Springer Link	https://link.springer.com/	19 December 2022	469
Emerald Insight	https://www.emerald.com/insight/	18 December 2022	23
IEEE Xplore	https://ieeexplore.ieee.org/	19 December 2022	57
ACM Digital Library	https://dl.acm.org/	10 December 2022	395
Google Scholar	https://scholar.google.com/	19 December 2022	530

**Table 3 sensors-23-02968-t003:** Body parts that receive the haptic stimuli.

Stimuli Location	Number of Papers	Papers
Whole body	2	[31,32]
Head	1	[33]
Ears	1	[34]
Neck	1	[23]
Back	2	[34,35]
Shoulders	1	[36]
Chest	2	[34,37]
Belly	1	[38]
Upper arms	3	[34,35,37]
Forearms	3	[39,40,41]
Wrist	4	[38,42,43,44]
Hand	5	[24,45,46,47,48]
Fingers	5	[24,47,49,50,51]
Upper legs	2	[34,35]
Lower legs	1	[52]
Variable/undefined	1	[53]

**Table 4 sensors-23-02968-t004:** Papers categorized by the number of places where the stimuli were applied.

Number of Stimuli Locations	Number of Papers	Papers
1	7	[31,32,38,43,45,46,53]
2	5	[36,37,48,50,51]
4	4	[33,34,42,52]
5	3	[23,35,49]
6–10	3	[24,41,44]
11+	3	[39,40,47]

**Table 5 sensors-23-02968-t005:** Papers grouped by the general purpose of the device.

Purpose	Number of Papers	Papers
Alerts	7	[23,32,36,42,45,51,53]
Experiencing music	6	[31,33,35,37,46,52]
Pitch perception	3	[24,40,48]
Learning	3	[39,49,50]
Dancing	2	[37,38]
VR	1	[34]
General sound perception	1	[43]
Driving directions	1	[44]
Watching visual media	1	[47]
Lip-reading aid	1	[41]

**Table 6 sensors-23-02968-t006:** Papers divided by the type of actuator used to generate haptic stimuli.

Actuator Type	Number of Papers	Papers
Non-specified vibration motor	6	[23,35,43,44,45,51]
Eccentric rotating mass motor	3	[24,38,40]
Linear resonant actuator	4	[33,34,42,47]
Piezoelectric vibration motor	1	[49]
Speakers	2	[35,46]
Subwoofer	1	[35]
Speaker driver	1	[36]
Bone conduction speaker	1	[52]
Tactile transducer	1	[31]
Solenoid coil	1	[50]
Voice-coil actuators	4	[37,39,41,48]
Smartphone vibration	1	[53]
Bumping robot	1	[32]

**Table 7 sensors-23-02968-t007:** Portability of devices presented in the papers.

Portability	Number of Papers	Papers
Portable	21	[23,24,33,34,36,37,38,39,40,41,42,43,44,45,47,48,49,50,51,52,53]
Non-portable	4	[31,32,35,46]

**Table 8 sensors-23-02968-t008:** Device specifications focusing on the input, its processing, and the output/stimuli application, as well as the number of users it was tested with.

			Signal Delivery			Testing
Paper	Audio Acquisition	Signal Processing	Number of Stimuli	Stimuli Application Method	Portable Device	Participants
Tan et al. 2020 [39]	N/A. Sounds from the International Phonetic Alphabet are represented, which are not gathered from the environment.	N/A	24	Forearm sleeve with a four-by-six tactor array that applies vibrations.	Yes	51
Sierra et al. 2021 [31]	Audio signal sent through a computer.	Frequency segregation and low-pass filtered white noise.	1	Bench that is globally excited by two transducers that vibrate the entire body of someone laying down on it.	No	10
Sakuma et al. 2017 [23]	Microphone array.	Speech recognition engine and direction of arrival estimation with cross-power spectrum phase method.	5	Semicircle-shaped device with five vibration motors worn on the back of the neck.	Yes	10
Alves et al. 2017 [35]	Audio file (.wav or.mp3).	Extraction of melodic information into MIDI and further filtering.	5	Bracelet and chair with four speakers on the backrest and a subwoofer on the seat that produce vibrations.	No	13
Sekiya et al. 2017 [32]	N/A. Sounds are not represented, but instead a robot hits the bed.	N/A	1	Rolling robot that bumps against a bed where someone is sleeping.	No	10
Chao et al. 2018 [38]	Music clip with raw signals of a music segment.	Beat extraction from signal peaks.	1	Vibration motor packaged either as a glove or as a belt that produces vibrations.	Yes	35
Perrotta et al. 2021 [42]	Microphone in the wrist device.	Filtering the frequencies with the greatest amplitudes in chunks of time.	4	Wristband with four vibratory motors.	Yes	18
Namatame et al. 2018 [49]	Mono audio signal file (5 ch).	Audio file was edited to play at 93 ± 2 dB.	5	Handheld device where each of the five fingers receives its own vibrations via piezoelectric elements.	Yes	26
Mirzaei et al. 2021 [34]	Sound sources from a virtual reality environment using Unreal Engine 4.	None specified.	4	Suit with fixed vibromotors on the chest and upper back, along thighs, upper arms, or ears.	Yes	20
Fletcher et al. 2020 [40]	Prepared audio file.	Pitch chroma analysis, which groups frequencies by octave to generate a spectral representation of relative pitch.	12	Device consisting of 12 vibration motors that apply the stimuli to the forearm.	Yes	12
Jain et al. 2020 [43]	Microphone in the wrist device.	None specified.	1	Vibration motor encased in a wristband.	Yes	12
Kim et al. 2018 [53]	Set of microphones placed in various locations of the home.	Acoustic features computation with noise reduction, stereo to mono sound conversion, vertical/harmonic separation.	1	Smartphone vibration.	Yes	14
DeGuglielmo et al. 2021 [33]	Microphone (Adafruit Max 9814).	Separation into lower and higher frequency components, key feature extraction: amplitude and frequency.	4	Headband with four vibration motors.	Yes	3
Carvalho et al. 2022 [46]	Audio samples created with virtual instruments using LOGIC X and Ableton Live.	N/A	1	Speakers that vibrate a table.	Yes	1
Otoom et al. 2022 [44]	Microphone.	Feature vector extraction and classification.	10	Two bracelets, one for each hand, each with five vibration motors.	Yes	10
Shimoyama 2021 [36]	Ear microphones (MDR-EX31BN, Sony).	Interaural time difference calculation.	2	One vibration motor taped to each shoulder.	Yes	1
Reed et al. 2022 [41]	Audio-visual recordings of consonants.	None specified.	7	Vibration device with seven tactors that is worn on the forearm.	Yes	1
Yağanoğlu and Köse 2018 [45]	Microphone.	Spectogram generation, peak finding, audio fingerprint hashing.	1	Standalone handheld vibration motor.	Yes	55
Shin et al. 2020 [24]	Microphone under a pop filter.	20 ms buffer, fast Fourier transformation for frequency data, peak frequency calculation.	9	Glove with nine vibration motors.	Yes	2
Ganis et al. 2022 [48]	Unfiltered audio from the smartphone.	Digital to analogue conversion and amplification.	2	Smartphone cover with two vibrating handles.	Yes	15
Cavdir 2022 [37]	Pitch-shifted audio files	None specified.	2	Two haptic actuators, each wrapped in a strap which allows the user to decide where to locate it.	Yes	1
Suzuki et al. 2017 [50]	N/A Mouse clicks (not the sounds) are represented as vibrations.	N/A	2	Two rings with a vibration motor each.	Yes	13
Garcia et al. 2022 [47]	N/A The haptic stimuli were designed for a video independent of sounds.	N/A	3	Glove with three vibrating coin motors.	Yes	16
Trivedi et al. 2019 [52]	Input from a musical keyboard through a MIDI connection.	Using LabView to ensure equal distribution of signals to the speakers	4	Pair of sleeves worn on the lower legs with two bone conduction speakers each.	Yes	5
Yağanoğlu and Köse 2017 [51]	Front, rear, left, and right microphones.	Z-Score normalization, feature extraction and classification.	2	Pair of vibration motors taped to the middle fingers.	Yes	56

## Data Availability

Not applicable.

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
