# Peer review of "Recent Developments in Haptic Devices Designed for Hearing-Impaired People: A Literature Review"

_sensors, 2023, doi:10.3390/s23062968_

Round 1

Reviewer 1 Report

The paper attempts to present a literature review of haptic technology vis-a-vis applications for hearing impaired. The authors use the PRISMA methodology for literature search which is fine, however it is not enough to just apply the methodology. It is also necessary to structure the results in a more readable fashion and comment, compare and synthesize various haptic systems and techniques used for such applications.

Also, many references are missing. For example, for a detailed survey on Haptic Interfaces please see:

Mark D. Fletcher (2021) Using haptic stimulation to enhance auditory perception in hearing-impaired listeners, Expert Review of Medical Devices, 18:1, 63-74, DOI: 10.1080/17434440.2021.1863782

F. G. Hamza-Lup, K. Bergeron, and D. Newton (2019) “Haptic Systems in User Interfaces: State of Art Survey,” In the Proceedings of the Association for Computing Machinery Southeast Conference, 18–20 April, Kennesaw, GA (pp.141-148). 

There is no significant contribution to the field however a more extensive review on haptic technology for hearing impaired structured in a better way (I suggest the use of table to compare various hardware & software system can improve the survey). The use of diagrams and other visual methods to compare various approaches is also a potential improvement.

A more detailed analysis of the literature review is necessary to improve the contribution to the field. 

Author Response

Thank you very much for your comments, that are greatly appreciated.

The paper attempts to present a literature review of haptic technology vis-a-vis applications for hearing impaired. The authors use the PRISMA methodology for literature search which is fine, however it is not enough to just apply the methodology. It is also necessary to structure the results in a more readable fashion and comment, compare and synthesize various haptic systems and techniques used for such applications.

Mark D. Fletcher (2021) Using haptic stimulation to enhance auditory perception in hearing-impaired listeners, Expert Review of Medical Devices, 18:1, 63-74, DOI: 10.1080/17434440.2021.1863782

Indeed we had referenced that same paper in “previous literature reviews”. Fletcher had divided his work in acquisition, processing and delivery. Our original idea was to focus on deli wey only, but we realized we could improve this section. Thus, we added also acquisition and processing. 

  1. G. Hamza-Lup, K. Bergeron, and D. Newton (2019) “Haptic Systems in User Interfaces: State of Art Survey,” In the Proceedings of the Association for Computing Machinery Southeast Conference, 18–20 April, Kennesaw, GA (pp.141-148). 

Our intention was to focus on papers presenting haptic devices that were tested on users. Therefore, these review papers were excluded, but mentioned in line 213.

There is no significant contribution to the field however a more extensive review on haptic technology for hearing impaired structured in a better way (I suggest the use of table to compare various hardware & software system can improve the survey).  

The work of Fletcher focuses on the analysis of audio acquisition, signal processing, and signal delivery. Table 7 includes these categories, and we hope that this will help visualize potential opportunities. For example, the table illustrates that vibrations are not the only stimuli that could be used for research. We can also see that portability is not an issue. Furthermore, we can observe that the location of the stimulus can vary.

The use of diagrams and other visual methods to compare various approaches is also a potential improvement.

We hope the addition of table7 will suffice.

A more detailed analysis of the literature review is necessary to improve the contribution to the field. 

We have also discussed contributions in greater detail. 

Reviewer 2 Report

The literature review presented in this paper is well conducted and the methodology seems appropriate. The manuscript is well written and organized. My only concern is about the citation life of the paper since ti comprehends a very specific interval (2017-2022).

-The methodology seems correct. Maybe the authors should include if the mentioned work considers experimental studies with volunteers and the number of volunteers in each case.

 -The quality of all figures could be improved if vectorial graphics are employed instead of bitmaps. Maybe augmenting the thickness of the arrows in Figure 1 could improve its presentation.  

-Please avoid the usage of short forms in English such as isn't -> is not, etc.

Author Response

The literature review presented in this paper is well conducted and the methodology seems appropriate. The manuscript is well written and organized. My only concern is about the citation life of the paper since ti comprehends a very specific interval (2017-2022).Due to the rapid evolution of hardware, our reasoning was that recent papers would contain the latest prototype innovations. In newer proposals, older prototypes would certainly have been considered. Even so, we included other literature reviews so that older papers were considered.

-The methodology seems correct. Maybe the authors should include if the mentioned work considers experimental studies with volunteers and the number of volunteers in each case. We have included table 7. We have also included a column that shows the number of participants in each study.

 -The quality of all figures could be improved if vectorial graphics are employed instead of bitmaps. 

We have replaced all JPGs for SVGs.

Maybe augmenting the thickness of the arrows in Figure 1 could improve its presentation.  

We enlarged the arrows and vectorized the images.

-Please avoid the usage of short forms in English such as isn't -> is not, etc.  We apologize. All informal language has been corrected.

Reviewer 3 Report

1. line 25, "levels.[3]." Please revise the punctuation correctly.

2. The authors should add 1 concept diagram in the discussion section to show the full picture of the future development scenario.

3. In the conclusion section, please highlight the contribution of this paper more.

Author Response

We appreciate and thank you for your comments.
1. line 25, "levels.[3]." Please revise the punctuation correctly.  It has been corrected.
2. The authors should add 1 concept diagram in the discussion section to show the full picture of the future development scenario. In the conclusion section, please highlight the contribution of this paper more.
We have discussed in greater detail the paper’s contributions.  We have also added table 7 that helps readers visualize areas of opportunity in each category.

Round 2

Reviewer 1 Report

Please update Table 1. Electronic databases and digital libraries searched for the literature review process (with December dates and numbers)

It would be better to organize table 7 in a more visually appealing way. Maybe a graph/diagram showing the numbers rather than a large, hard to follow table.  Also please emphasize the main conclusion from Table 7. The Type of Stimuly and Portable Device columns are redundant as all have the same value.

Please have an English speaking person review the language as there are many ways to improve the statements. 

Author Response

Once again, thank you for your comments. The questions will be answered one by one.

  • Please update Table 1. Electronic databases and digital libraries searched for the literature review process (with December dates and numbers)

Table 1 has been updated with the most recent search results. During the past two weeks, most searches remained the same. Even though the most recent paper was excluded by the criteria, we mentioned it in the introduction.

  • It would be better to organize table 7 in a more visually appealing way. Maybe a graph/diagram showing the numbers rather than a large, hard to follow table.  Also please emphasize the main conclusion from Table 7. The Type of Stimuly andPortable Device columns are redundant as all have the same value.

Table 7 has been modified to eliminate the columns The Type of Stimuli and Portable Device, and the results have been summarized. A new graph shows the part of the body where stimuli were applied. We used a heatmap to show how many stimuli were applied to each region. Hopefully, this figure will help researchers identify areas of opportunity for further research on stimuli.

  • Please have an English speaking person review the language as there are many ways to improve the statements. 

We apologize for the poor quality of our English, since it is not our native language. A native-speaking style editor had corrected the original script, but we had not corrected the changes we made from the original paper. We have now modified these paragraphs. I also hope you will find this new version satisfactory.

Finally, we highlighted Table 7's key findings.

Reviewer 3 Report

The author responded well to all my suggestions.

Author Response

Thank you very much. We included Table 7, a new figure and changes to both discussion and conclusions, based on suggestions by another reviewer.